# Prevalence of poor glycemic control and the monitoring utility of glycated albumin among diabetic patients attending clinic in tertiary hospitals in Dodoma, Tanzania: A cross-sectional study protocol

**George Gabriel Mkumbi**[1,2]*, **Matobogolo Boaz**[1,2,3]

1 Department of Internal Medicine, School of Medicine and Dentistry, The University of Dodoma, Dodoma, Tanzania, 2 Department of Internal Medicine, Benjamin Mkapa Hospital, Dodoma, Tanzania, 3 Department of Internal Medicine, University of Dodoma Hospital, Dodoma, Tanzania

* mkumbig@gmail.com

**Data Availability Statement:** Since this publication is a Study Protocol for a proposed study, no

## Abstract

The burden of diabetes is rising in developing countries, and this is significantly linked to the increasing prevalence of poor glycemic control. The cost of glycated haemoglobin (HbA1c) testing is a barrier to timely glycemic assessments, but newer tests such as glycated albumin may be cheaper and tempting alternatives. Additional research must ascertain if glycated albumin (GA) can act as a viable supplement or alternative to conventional HbA1c measurements for glycemic control in diabetic individuals. GA as a biomarker is an emerging area of interest, particularly for those who display unreliable HbA1c levels or cannot afford the test. This study aims to investigate the prevalence of poor glycemic control in outpatient diabetic patients and the utility of glycated albumin in this population's monitoring of glycemic control. **Method.** A cross-sectional study of 203 diabetic patients will be conducted at the Dodoma Regional Referral Hospital and Benjamin Mkapa Hospital from August 1st, 2023, to August 31st, 2024. Patients diagnosed with diabetes mellitus for over six months will be screened for eligibility. Informed consent, history, clinical examination, and voluntary blood sample collection will be obtained from all eligible patients. Glycated Albumin levels will be obtained from the same blood samples collected. The glycemic status of all patients will be defined as per HbA1c, and a level of greater than 7% will considered as a poor control. The analysis will be computed with SPSS version 28.0, and a predictor variable, $P<0.05$, will be regarded as statistically significant, with the utility of GA determined by plotting the area under the ROC curve and the confusion matrix.

## Introduction

Diabetes mellitus has been one of the twenty-first century's most serious healthcare challenges [1]. The disease is spreading at an alarming rate worldwide, with developing nations

datasets were generated or analysed during the current publication. All relevant data from this study will be made available upon study completion in subsequent publications.

**Funding:** The authors received no specific funding for this work.

**Competing interests:** The authors have declared that no competing interests exist.

accounting for 80% of the new cases [2]. In 2015, it was reported to affect 8.8 per cent (415 million) of adults worldwide, and it is expected that 652 million individuals (10.4 per cent) will have diabetes by 2040 [3]. Tanzania, like the rest of Sub-Saharan Africa, has a high diabetes burden, with rising prevalence (14.8%), complications, and death, as well as life-threatening impairments [4, 5]. According to the International Diabetes Federation (IDF), an estimated 400,000 people are living with diabetes in the country [6]. Poor glycemic control can lead to diabetic complications, which is a significant concern in diabetes management.

According to the American Diabetes Association (ADA), only 50% of patients globally achieve adequate glycemic control [7]. In sub-Saharan Africa (SSA), glycemic control rates remain suboptimal [8]. In Tanzania, recent studies have reported high rates of poor glycemic control, ranging from 67.7% to 73.8% [6, 8–10]. Managing diabetes in SSA presents many challenges, including resource inadequacies, competing traditional healthcare priorities, limited preparedness for chronic disease management, and low health insurance coverage [6, 9]. Addressing diabetes control is of paramount urgency to alleviate the mounting burden of this disease in the sub-Saharan African region. Formulating and implementing effective diabetes control strategies necessitates a comprehensive understanding of the multifaceted factors contributing to glycemic control, enabling the identification of targeted interventions [6].

To limit the risk of diabetes complications, diabetic patients must be appropriately diagnosed and monitored [11]. Diabetes diagnosis and prognosis primarily depend on two tests: serum/blood glucose and glycosylated haemoglobin (HbA1c) [12]. These metrics, however, are not failsafe, and their therapeutic utility is influenced by various clinical and analytical parameters [13]. Other glucose homeostasis markers, such as fructosamine and glycated albumin (GA), may be viewed as a desirable alternative, particularly in individuals whose HbA1c test results are inaccurate. For instance, individuals with hemoglobinopathies and kidney illness exhibit rapid alterations in glucose homeostasis and more excellent glycemic excursions [14, 15]. According to existing information, GA appears to have a higher overall diagnostic efficiency than fructosamine in various clinical contexts [16].

GA is a glycemic control indicator examined as a substitute for HbA1c in people with diabetes mellitus. GA is a far more dependable glycemic variability indicator than HbA1c. Accumulating evidence suggests that GA is a precise diagnostic test directly related to microvascular complications in diabetes [13]. Additionally, it is suitable for individuals undergoing hemodialysis, and its levels are unaffected by anaemia or haemolytic processes. GA is preferable to fructosamine because different serum proteins do not impact it differently. The catalytic technology used to analyse it is easy to deploy and quick to execute, in addition to being highly effective analytically and possessing greater standardization [17]. In clinical situations where HbA1c values are erroneously altered, GA testing may provide a reliable result for monitoring DM, according to a recent study. This is because the physiological mechanisms by which these two glycated proteins are produced guarantee that GA outperforms HbA1c in assessing glucose homeostasis without confounding factors [13].

Additional advantages of glycated albumin over glycosylated haemoglobin include decreased reagent costs and the possibility of automating glycated albumin measurement on numerous conventional laboratory instruments [18]. Although additional research is required to evaluate whether GA could augment or even replace traditional diabetic markers such as HbA1c, GA may assist in the diabetic care of patients with inaccurate HbA1c readings [19]. As a final regard to the benefits mentioned earlier of GA, an international consensus on clinical usage is required to ensure its inclusion in routine clinical laboratory workup, hence improving future monitoring and management of DM patients. Much similar research, like the one proposed, is required for this to happen.

## Materials and methods

### Study aims

1. To determine the prevalence of poor glycemic control among diabetic patients attending clinics in Dodoma Tertiary hospitals.

2. To determine the utility of serum glycated albumin as an index of glycemic control in diabetic patients attending clinics in Dodoma Tertiary hospitals.

### Study design

For twelve months, A cross-sectional study will be conducted at the Benjamin Mkapa Hospital and the Dodoma Regional Referral Hospital in Dodoma, Tanzania.

### Study setting

The medical centres at both BMH and DRRH will serve as the study sites. Dodoma, the capital city of Tanzania, is home to several medical facilities. Dodoma Regional Referral Hospital (DRRH) and Benjamin Mkapa Hospital (BMH) are both referral hospitals for the Dodoma region, the central zone, and surrounding areas, respectively. In addition, the University of Dodoma uses both hospitals as teaching centres (UDOM). These facilities partly facilitate health care for those living in the central zone. The population of the Dodoma region was 2.49 million, as per the 2012 Population and Housing Census.

BMH and DRRH have a capacity of 500 and 420 beds, respectively. Each month, approximately 1400 patients with medical-related diagnoses are seen in the medical clinics at BMH, with fifteen to twenty (15–20) of these patients having diabetes mellitus being seen every day. Additionally, 480 patients with diabetes are documented at DRRH each month. Thus, a total of 10–15 patients with diabetes mellitus are typically seen per clinic session, and a total of 30–35 patients per session from both health facilities combined (Unpublished data).

### Sample size estimation

Following the Leslie-Kish formula [14, 15], the minimum sample size required to include a patient in the study will be determined as below;
Where:

$$N = \frac{Z^2 p(1-p)}{e^2}$$

N = Minimal sample size
Z = The value from the normal distribution scale that indicates a degree of significance (1.96 for 95% confidence level).
p = 84.3% is the rate of uncontrolled blood sugar among diabetic patients who visit the outpatient clinic [20]
e = margin of error (usually 5%) = 0.05
So;

$$N = \frac{1.96^2 \times 0.84 \times (0.16)}{0.05^2}$$

N = 203.38 = 203
*Therefore, the total sample size will be 203 participants*

## Inclusion criteria

1. Patients with at least 18 years of age.

2. Patients who will accept participation in the study through signing an informed consent.

3. Patients on oral antidiabetic drugs (OAD), insulin, or a combination of the two for at least six months.

## Exclusion criteria

1. Patients who have received blood transfusions during the last three months.

2. Patients who are receiving erythropoietin.

3. Patients who are receiving Iron supplements.

## Participants characteristics

The study participants will be adults (18 years and older) who have been diagnosed with diabetes mellitus and are attending the diabetic clinics at Benjamin Mkapa Hospital and Dodoma Regional Referral Hospital during the study period.

## Sampling technique

In this study, the consecutive sampling method will be used to recruit the participants. This will enable the researcher to target and track the available patients as they visit the clinic. Therefore, the sampling technique will allow the researcher to contact any patient meeting the criteria.

## Data collection process

The data collection process will begin in August 2023 until January 2024. Three research assistants will be involved and trained by the principal investigator to guarantee that the questionnaires are filled out correctly and that no research misconduct occurs during data collection. Two certified nurses and one laboratory technician will serve as researcher assistants. Each assistant interviewer will be trained to ask participants the same pertinent questions listed in the questionnaire to minimise the researcher's bias during data collection. Respondents will be screened before the start of the clinic, and the questionnaires will be completed after informants have received service. The respondents' names will be removed from the questionnaires, and a coding system will be used to represent their information throughout the data processing process. All responses will be kept strictly confidential and utilised for academic analysis.

**Participant interview.** A participant who provided informed consent will be interviewed and demographic data such as age, gender, and level of education will be collected. In addition, a minimum of two contacts, one of the patients and the second one of the next of kin, will be recorded. As part of social demographic data, alcohol consumption history (defined as alcohol consumption within the past 12 months), with an emphasis on duration and frequency, will be

collected [21, 22]. In addition to the above duration and number of cigarettes in the number of packs per day, the history of current smoking, defined as smoking within the past year, will be obtained [21]. A history of hypertension (defined as a history of hypertension or the use of antihypertensive medications) and diabetes type (defined as a history of diabetes or the use of diabetic medications) will be documented [23, 24].

**Laboratory procedure.** Respondents will be asked to consent to a venepuncture by a trained phlebotomist at each centre. Three milliliters of blood will be drawn on either side of the cubital fossa using a 5cc syringe and placed in a red and lavender top vacutainer tube. The phlebotomist will be trained on how to handle any adverse reaction to venepuncture if they arise. The blood sample for both study blood collection centres will be transported to the authorised Benjamin Mkapa Hospital (BMH) laboratory in a refrigerated box with an ice pack for planned tests. Blood samples will be processed according to the BMH laboratory's Standard Operating Procedures for each test.

## Data collection tools

This study will use standardized, structured surveys with closed and open-ended questions to collect data as described above. The questionnaire will consist of three parts: the first will assess the patient's sociodemographic characteristics; the second will evaluate the patient's clinical characteristics; and the third will include laboratory results and bio data like the patient's glycated haemoglobin and haemoglobin A1c levels, fasting blood glucose, and random blood glucose levels.

If respondents choose to complete the surveys independently, we will provide them with a Swahili translation. Enzymatic determination of GA will be performed using a liquid reagent (Lucica GA-L® Asahi Kasai Pharma Co., Tokyo, Japan) that consists of ketoamine oxidase and an albumin-specific proteinase, and the results will be evaluated using a Cobas 6000 series HPLC analyser. Automated high-performance liquid chromatography from the National Glycated Haemoglobin Standard Program and the International Federation of Clinical Chemistry will calculate the HbA1c (per cent) value.

## Validity of the data collection tools

Validity will be defined as the degree to which tools or tests accurately measure what should be measured [25]. The standardized equipment and structured questions used in this study will be re-evaluated to ensure they are appropriate for the local situation and adequately assess the required variables. The draft version of the structured instrument and questions for measuring the variables will be provided to specialists from the University of Dodoma's School of Medicine and Dentistry and the supervisors. Modifications will be made in consultation with specialists and the supervisor. During the pilot project, data-gathering tools will be pre-tested.

## Reliability

The term "reliability" refers to the state of being accurate or precise and the consistency of the data gathered during the investigation [25]. The glycated haemoglobin test is considered the gold standard for monitoring glucose levels, and its sensitivity and specificity will serve as the basis for this study's reliability (HbA1c). Additionally, structured questionnaires will be used to elicit information from informants. To guarantee the data's reliability, the study's tool will be pre-tested to determine its correctness and ability to produce the desired results.

## Pilot study

A pilot study is defined as a smaller-scale version of a larger-scale study conducted to improve and clarify aspects of the study's methodology, such as the tools and processes to be utilised in data gathering will be performed [26]. A pilot study aims to determine whether or not the full-scale investigation is feasible and to reveal study strengths and flaws. This study performed field testing with 10% of the minimal sample size at DRRH in mid-July. The pilot study specifically tested for adequacy, feasibility, and estimation of the time required for filling questionnaires to identify the problems that may occur during research. The information obtained from the pilot study is hoped to facilitate conducting the proposed study.

## Measurements of the variables

**1. Dependent variables.**   In this study, we will assess the serum levels of glycosylated haemoglobin (HbA1c) and glycated albumin (GA), biomarkers for determining the glycemic status in individuals with diabetes. These markers as variables will be considered the study's dependent variables and normative values will be employed in their evaluation.

**2. Independent variables.**

i. *Demographic variables*. Sociodemographic variables such as respondent age, gender, marital status, education, and insurance status are determined as independent variables in this study. A nominal scale will classify participants according to sex, insurance status, and marital status. Age and educational background will be presented in ordinal scales.

ii. *The patients' clinical characteristics*. Patients' clinical profiles, such as commodity, prescription, drug adherence, and duration of illness, all have a marked influence on glycemic control and, therefore, are among the independent factors.

## Data analysis plan

After data entry into the master coding sheet is complete, it will be analysed using SPSS version 28.0. The data will be cleaned and completeness checked by calculating the frequency of all variables. Descriptive statistics will be utilised to examine the demographics of the respondents. The percentage and median ages will be published, and the percentages of each gender, place of residence, degree of education, and marital status will also be shown. The percentage of those with poor glycemic control will be calculated by dividing the number of respondents with increased HbA1c values by the total number of respondents who had the test.

The findings will be shown in the form of a count (and percentage), a mean (and standard deviation), and a median (25th-75th percentiles). Comparisons of baseline characteristics between glycemic-control subgroups will be made using chi-square, analysis of variance, and Kruskal-Wallis tests. The correlation between two continuous measures will be estimated using the covariance estimate of the multivariate t distribution. This will provide some resistance to extremes without an excessively high breaking threshold. Using a single dependent variable, two correlation coefficients will be calculated and the significance of their difference will be tested using the Williams test. Segmented regression analysis will be used to examine the ties between indicators of glucose homeostasis.

This regression method generates separate regression coefficients for possible piecewise linear relations. The split point between two segmented relations will be calculated using the outcomes of Davies' test for a non-zero difference in slope between variables. GA % and HbA1c's ability to predict the existence of impaired glucose tolerance will be assessed and compared

using the area under the receiver operating characteristic curve (AUC, C-statistic). Youden's J-point will be used to establish diagnostic cut-offs for the different types of glucose intolerance. The Kappa statistic will be used to measure the degree of agreement between markers at these arbitrary cut-off points, and the 95% CI will be determined using 2000 replicates with stratified bootstrap and percentile approaches. These cut-offs will be used to assess the performance of GA per cent and HbA1c using the following performance measures (with a 95% confidence interval): sensitivity, specificity, Youden's Index, positive predictive value (PPV), negative predictive value (NPV), accuracy, diagnostic odd ratio (DOR), number needed to diagnose (NND), the likelihood ratio of a positive test (LR+), and likelihood ratio of a negative test (LR-). Assuming parallel testing, we will assess the diagnostic utility of combining GA % and HbA1c. If the p-value is less than 0.05 on two-sided testing, the result is considered statistically significant with a 95% CI. Cronbach's alpha will be used to check the data's reliability. The study will report its findings by the Standards for the Reporting of Diagnostic Accuracy Studies (STARD).

## Ethical issues

This study will follow ethical guidelines throughout the project, from participant recruitment to data collection through analysis to dissemination of results. Participants will be briefed about the study's nature and goals and will be able to revoke their permission at any moment during the study, and will get the same level of treatment as those who choose to participate. The results of this study will be disclosed for the sole purpose of formulating an action plan. The University of Dodoma (UDOM) institutional research review ethics committee has blessed the PI's request for ethical approval through a letter with Ref No. MA.84/261/02/9. Permission to collect the data was granted in both hospitals through a letter With Ref No. DB.122/467/01G/29 And GA.244/292/01/'E'/45 respectively. The researcher will alert the proper authorities, including attending clinicians, to follow up with respondents with laboratory findings requiring quick care.

## Study timeline

This study will last for twelve calendar months, from 1st August 2023—six months of data collection, four months of data analysis, and two months for manuscript write-up.

## Discussion

Glycemic monitoring is an essential part of diabetes therapy, and glycated albumin (GA) and glycated haemoglobin (HbA1c) are two typical indicators [27]. Although both help track how healthy diabetes is managed, they are measured and used clinically in different ways. Whereas HbA1c represents glucose levels on average over the previous two to three months, the usual lifespan of a red blood cell [28]. Haemoglobin beta chain glycosylation occurs when glucose covalently bonds to the amino acid valine at the N-terminal end of the beta chain [29]. Several guidelines for diagnosing and managing diabetes advocate measuring HbA1c as a benchmark [30]. Conversely, GA is generated when glucose binds to albumin without the help of an enzyme and hence represents glycemic control during the previous two to three weeks [31]. GA forms at a much quicker rate than HbA1c [32].

HbA1c and GA can be used as indicators of glycemic control, but each has unique clinical applications [17]. There is a high correlation between HbA1c levels and the risk of microvascular and macrovascular problems in diabetes, and HbA1c has been widely utilised as a marker of glycemic control in the disease [29]. The American Diabetes Association (ADA) recommends that most individuals with diabetes aim for a HbA1c of 7%, with a target of 6.5% for

some patients [33, 34]. However, there are specific clinical contexts where HbA1c cannot be reliably used. HbA1c readings may be misleading, for instance, in those with hemoglobinopathies or anaemia. Haemolysis, blood transfusion, and other red blood cell turnover disorders can also affect HbA1c levels [30, 35, 36]. However, liver illness, pregnancy, and other diseases that modify albumin turnover may affect GA [36]. While HbA1c can be impacted by factors like erythrocyte turnover and drugs that shorten or lengthen the lifespan of erythrocytes, GA is unaffected by these things, and its measurement is less prone to inconsistency (6). Chronic kidney disease (CKD) patients may also benefit from GA [30, 35]. Falsely low HbA1c values in CKD patients may be caused by erythropoietin insufficiency, anaemia, and erythrocyte fragmentation [37–41]. On the other hand, GA is not influenced by these factors and may be a more reliable indicator of glycemic control in CKD patients [36].

Several studies have shown that GA levels are linked to HbA1c levels and the risk of microvascular and macrovascular complications in diabetes [32]. One study indicated that GA levels were substantially related to the danger of diabetic eye disease and kidney disease [42]. Additional research has linked elevated GA levels to type 2 diabetics' risk of developing and experiencing severe coronary artery disease [32, 36]. A recent study also suggested that GA, rather than HbA1c, may be a more accurate predictor of cardiovascular events in those with type 2 diabetes [43].

When comparing glycated albumin to HbA1c for diabetes control, a cross-sectional study has various limitations. Uncertainty over timing, potential for bias in selection or measurement, and restricted applicability are potential limitations. Care has been taken to address these drawbacks in multiple ways, such as reducing Selection bias with careful sampling and correcting measurement bias with tried-and-true methods detailed earlier. The findings are usually more broadly applicable if they are based on a diverse and representative study population. Hence, two centres that serve a diverse group of diabetic patients from the nation's capital were chosen. These methods aim to improve the trustworthiness and validity of the findings.

In conclusion, glycated haemoglobin (HbA1c) is the current standard marker for monitoring glycemic control, which is critical in managing diabetes [23, 44–46]. GA may offer some benefits over HbA1c in specific therapeutic settings despite HbA1c being a well-known marker with standardized measurement and documented clinical importance [23]. With a shorter half-life, more consistent measurement, and immunity to confounding factors, including erythrocyte turnover and blood transfusion, GA shows promise as a helpful marker of glycemic control in individuals with diabetes [33]. This is especially true for people whose glycemic control is less stable or who have chronic renal disease. Despite GA not being as frequently investigated as HbA1c, it has been linked in several studies to microvascular and macrovascular problems in diabetes. Therefore, the current study aims to show the actual therapeutic utility of GA and its applicability to our settings.

## Supporting information

**S1 Table. Table showing assessment of Aim 1.**
(TIF)

**S2 Table. Table showing assessment of Aim 2.**
(TIF)

## Acknowledgments

The authors would like to thank the participants, staff of Dodoma Regional Referral Hospital, and Benjamin Mkapa Hospital for donating their time to this project and the Ministry of Health.

## Author Contributions

**Conceptualization:** George Gabriel Mkumbi, Matobogolo Boaz.

**Data curation:** George Gabriel Mkumbi, Matobogolo Boaz.

**Formal analysis:** George Gabriel Mkumbi, Matobogolo Boaz.

**Funding acquisition:** George Gabriel Mkumbi, Matobogolo Boaz.

**Investigation:** George Gabriel Mkumbi, Matobogolo Boaz.

**Methodology:** George Gabriel Mkumbi, Matobogolo Boaz.

**Project administration:** George Gabriel Mkumbi, Matobogolo Boaz.

**Resources:** George Gabriel Mkumbi, Matobogolo Boaz.

**Software:** George Gabriel Mkumbi, Matobogolo Boaz.

**Supervision:** Matobogolo Boaz.

**Validation:** George Gabriel Mkumbi, Matobogolo Boaz.

**Visualization:** George Gabriel Mkumbi, Matobogolo Boaz.

**Writing – original draft:** George Gabriel Mkumbi, Matobogolo Boaz.

**Writing – review & editing:** George Gabriel Mkumbi, Matobogolo Boaz.

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
