## [Decision Letter · Decision Letter 0]

27 Sep 2023

PONE-D-23-20966Prevalence and monitoring utility of glycated albumin among diabetic patients attending clinic in tertiary hospitals in Dodoma, Tanzania: A cross-sectional study protocolPLOS ONE

Dear Dr. Mkumbi,

Thank you for submitting your manuscript to PLOS ONE. After careful consideration, we feel that it has merit but does not fully meet PLOS ONE’s publication criteria as it currently stands. Therefore, we invite you to submit a revised version of the manuscript that addresses the points raised during the review process.This manuscript requires a major revisionConsider registering the study in one of clinical studies data bases like clinicaltrial.govThoroughly address all comments raised by reviewers==============================

We look forward to receiving your revised manuscript.

Kind regards,

Fredirick Lazaro mashili, MD, PhD

Academic Editor

PLOS ONE

Additional Editor Comments:

The authors present a protocol of a planned study aimed at determining the prevalence and monitoring utility of glycated albumin among diabetic patients attending clinic in tertiary hospitals in Dodoma, Tanzania. The study is relevant, and the document provides a clear outline of the study's intent, methodology, and expected outcomes. The research seems well-thought-out, with comprehensive planning around data analysis and ethical considerations. Enhancements can be made in clarifying certain sections and ensuring that all potential limitations and implications are thoroughly addressed as follows.

1. A range of statistical techniques have been listed, indicating a comprehensive plan to thoroughly analyze the data. However, the authors have mentioned calculating a percentage for demographic variables (e.g., gender, place of residence). They should be cautious with smaller sample sizes as percentages may not provide a true representation. Actual counts or proportions might be more informative in such cases.

2. The inclusion of Cronbach's alpha to assess data reliability is a strong addition.

3. The research plan has clearly laid out the ethical considerations and approvals needed. It's good to see the references to consent and the process for withdrawing from the study. For completeness and clarity, the authors should consider having a plan to handle any adverse events or complications that might arise from the blood draw.

4. The timeline is straightforward. However, depending on the sample size, two months for data collection and only one month for analysis might be tight, especially if unforeseen issues arise. If that is the case the authors can consider amending their protocol based on realistic timelines and add in the protocol some of the comments raised here.

5. Comparison of GA and HbA1c is an essential part of this work. Ensure that the benefits and limitations of each are clearly demarcated and not mixed. Additionally, while the authors have addressed the clinical implications of HbA1c and GA, it may be worth briefly mentioning any cost or accessibility differences between the two, if relevant.

6. It's commendable that they have addressed potential study limitations, but consideration to adding a dedicated "Limitations" section will improve clarity.

7. The conclusion neatly sums up the relevance of the study. However, considerations to add potential implications for policy or clinical practice based on expected findings will add value.

8. The conclusion mentions the "current gold standard marker." However, a reader unfamiliar with the topic might benefit from clarifying that HbA1c is that standard marker.

9. Consider adding a statement about the potential for presenting findings at conferences or symposiums, if applicable.

10. While the hospitals and participants have been acknowledged the authors should also consider acknowledging any funding or grants that supported this work, if applicable.

11. Consider registering the study to one of clinical research data bases like clinical trial.gov

12. Lastly, they should ensure a thorough proofreading for grammatical and typographical errors before the final submission.

Reviewers' comments:

Reviewer's Responses to Questions

**Comments to the Author**

1. Does the manuscript provide a valid rationale for the proposed study, with clearly identified and justified research questions?

Reviewer #1: No

Reviewer #2: Yes

2. Is the protocol technically sound and planned in a manner that will lead to a meaningful outcome and allow testing the stated hypotheses?

Reviewer #1: Yes

Reviewer #2: Yes

3. Is the methodology feasible and described in sufficient detail to allow the work to be replicable?

Reviewer #1: Yes

Reviewer #2: Yes

4. Have the authors described where all data underlying the findings will be made available when the study is complete?

Reviewer #1: No

Reviewer #2: No

5. Is the manuscript presented in an intelligible fashion and written in standard English?

Reviewer #1: Yes

Reviewer #2: No

6. Review Comments to the Author

You may also provide optional suggestions and comments to authors that they might find helpful in planning their study.

Reviewer #1: 1. The manuscript is written in the future tense while the study has already been conducted, additionally the manuscript reads like a research proposal.

2. Although there is a discussion section there are no results to be discussed.

3. Manuscript arrangement needs to follow standard journal format

4. Results and conclusion seem to be missing from abstract

5. Key references are needed, for example: “Additionally, it is suitable for individuals’ undergoing hemodialysis and its levels are unaffected by anaemia or haemolytic processes. GA is preferable to fructosamine because it is not impacted differently by different serum proteins”

Reviewer #2: GENERAL COMMENTS

1. I would advise thoroughly checking any typos and grammatical problems throughout the study protocol. This will make the proposal more comprehensible and professional-looking. For instance, glycosylated and glycated hemoglobin are mixed in various locations.

2. There are some repetitions of similar explanations in the paragraphs. This repetition can be confusing for the reader and can make the study protocol not appealing and seem less polished. Authors should thoroughly revise the whole document and make sure they remove/paraphrase wordy sentences and descriptions recurring in different sections. For example, The explanations about data collection by the use of questionnaires have been repeated several times in the text.

3. Why were other factors that may affect glycemic control like physical activity, and diet not considered.?

4. Authors should define all abbreviations the first time they are used. Also, abbreviations should be used consistently throughout the study proposal.

5. Under the Discussion section, authors should revise thoroughly and provide a reference for the applicability of cross-sectional studies in studying the Validity of diagnostic tests.

SECTIONS COMMENTS.

1. Introduction

Add a section describing the prevalence of poor glycemic control in Tanzania

2. Sample size estimation.

Can the authors provide any justification for using the cited reference from Uganda with a prevalence of 84.3%? There are a number of published studies conducted in Tanzania with a lower prevalence of poor glycemic control which would be a better choice than the one used.

3. Exclusion and Inclusion criteria

How will you identify these patients? “Patients who are at risk of receiving or donating blood”

Will the study not consider specific types of diabetic patients? E.g Type 1, type 2, autoimmune diabetes e.t.c?

4. Clinical examination

The names, brands and models, and city of manufacture of the equipment and machines should be written in brackets. For example, you write: "A blood pressure (BP) reading will be taken using an automated digital machine (AD Medical Instruments, Beijing, China)." Revise this throughout the manuscript.

5. Data collection methods.

Revise the dates for the commencement of data collection and, pilot study

In this section, it is stated that BP will be measured using a Mercury sphygmomanometer while in contrast to the previous section (clinical examination), it is clearly explained that BP will be gauged using an automated digital BP machine. To make it clear why is this so?

6. Independent variables

Revise the paragraph describing Glycemic control. what you describe is not related to the description of glycemic control as the independent variable. The paragraph is poorly organized, incoherent and congested with unnecessary explanations.

7. Ethical issues

Give a concise elaboration. Don't repeat explanations from the methodology sections.

8. Discussion

Provide a reference for the use of HbA1c a Gold standard method. A number of studies recommend OGTT as the Gold standard method.

Add more limitations of the study.

9. Dissemination of the findings

The statement “different peer-review journals for publication” should be revised. Duplicate or Salami publication is unethical in research. Manuscript should be submitted to only one peer-reviewed journal. After the previous submission's editorial decision has been made, resubmissions are permitted. .

7. PLOS authors have the option to publish the peer review history of their article (what does this mean?). If published, this will include your full peer review and any attached files.

Reviewer #1: No

Reviewer #2: **Yes: **Oscar Mbembela

---

## [Author Response · Author response to Decision Letter 0]

11 Nov 2023

Specifically, I would like to highlight the following fundamental changes that have been made:

1. I have revised the title to enhance clarity to read as written in the subject of this letter titled response to reviewers.

2. Table 1 to 4 below Details the significant revisions and how they address the reviewers' comments in the same document.

Kindly find the attached document.

---

## [Decision Letter · Decision Letter 1]

24 Jan 2024

PONE-D-23-20966R1PREVALENCE OF POOR GLYCEMIC CONTROL AND THE MONITORING UTILITY OF GLYCATED ALBUMIN AMONG DIABETIC PATIENTS ATTENDING CLINIC IN TERTIARY HOSPITALS IN DODOMA, TANZANIA: A CROSS-SECTIONAL STUDY PROTOCOLPLOS ONE

Dear Dr. Mkumbi,

Thank you for submitting your manuscript to PLOS ONE. After careful consideration, we feel that it has merit but does not fully meet PLOS ONE’s publication criteria as it currently stands. Therefore, we invite you to submit a revised version of the manuscript that addresses the points raised during the review process.

Please address the comments raised by one of the reviewersThis manuscript requires a minor revisionPlease format your manuscript accordingly 

We look forward to receiving your revised manuscript.

Kind regards,

Fredirick Lazaro mashili, MD, PhD

Academic Editor

PLOS ONE

Journal Requirements:

Additional Editor Comments:

Please address the comments raised by one of the reviewers

Reviewers' comments:

Reviewer's Responses to Questions

**Comments to the Author**

1. Does the manuscript provide a valid rationale for the proposed study, with clearly identified and justified research questions?

Reviewer #2: Yes

Reviewer #3: Yes

2. Is the protocol technically sound and planned in a manner that will lead to a meaningful outcome and allow testing the stated hypotheses?

Reviewer #2: Yes

Reviewer #3: Yes

3. Is the methodology feasible and described in sufficient detail to allow the work to be replicable?

Reviewer #2: Yes

Reviewer #3: Yes

4. Have the authors described where all data underlying the findings will be made available when the study is complete?

Reviewer #2: Yes

Reviewer #3: Yes

5. Is the manuscript presented in an intelligible fashion and written in standard English?

Reviewer #2: Yes

Reviewer #3: Yes

6. Review Comments to the Author

You may also provide optional suggestions and comments to authors that they might find helpful in planning their study.

Reviewer #2: Extensive revisions have been made by the authors as it was suggested. I would therefore recommend accepting the manuscript for publication

Reviewer #3: REVIEWER COMMENTS

The authors intend to determine the prevalence of poor glycemic control and the utility of serum glycated albumin (GA) as an index of glycemic control in diabetic patients. The proposed study is important since GA appears to be of great potential in diagnostic efficiency, relatively less costly and the possibility of GA equipment automation. In turn, GA as biomarker for glycemic control will be useful in limited resource settings such as Tanzania.

Nevertheless, the authors need to address the following comments;

1. It is not clear why the authors will screen for eligibility among patients diagnosed with diabetes mellitus for over six months. What was the basis for using 6 months as a cut-f point for eligibility to participate in the study?

2. The authors intend to conduct an interview and self-administer the questionnaire among those would wish to do so. In addition, those wishing to self-administer the question will be given a Swahili version questionnaire. How is this likely to affect the study findings?

3. The authors describe that they will perform physical examinations and/or obtain information such as blood pressure, body mass index and electrocardiogram. However, the relationship between such parameters and the validity of GA in glycemic control is not clearly described in the introduction and methods section such as in the analysis section.

7. PLOS authors have the option to publish the peer review history of their article (what does this mean?). If published, this will include your full peer review and any attached files.

Reviewer #2: **Yes: **Oscar Mbembela

Reviewer #3: No

---

## [Author Response · Author response to Decision Letter 1]

21 Mar 2024

comment 1: It is not clear why the authors will screen for eligibility among patients diagnosed with diabetes mellitus for over six months. What was the basis for using 6 months as a cut-off point for eligibility to participate in the study?

response1 : • The six-month period has no meaning and was chosen arbitrarily just to exclude new and unstable diabetic patients from participating in the study while targeting stable anti-diabetic medication clients that have been at least on medication for some time and are assumed to have a set regimen that they adhere to. This aligns with the goal of the study to estimate glycemic control and suggest a better test, rather than to diagnose the disease. 

comment 2: The authors intend to conduct an interview and self-administer the questionnaire among those would wish to do so. In addition, those wishing to self-administer the question will be given a Swahili version questionnaire. How is this likely to affect the study findings?

response 2: • It is the author's view that the questionnaires to be as simple and clear that an individual with basic reading and writing can understand and fill them accurately with no or minimal assistance, in a language that is native and common in the study area. It is our view that when this chance is granted it improves participation rates and lowers the data collection burden. With proper proofreading and review of all submissions, it's unlikely to affect the results finding.

comment 3: The authors describe that they will perform physical examinations and/or obtain information such as blood pressure, body mass index, and electrocardiogram. However, the relationship between such parameters and the validity of GA in glycemic control is not clearly described

response 3: • The author acknowledges the fact that this was an unnecessary undertaking and it is removed from the methodology section.

---

## [Decision Letter · Decision Letter 2]

19 Jun 2024

PREVALENCE OF POOR GLYCEMIC CONTROL AND THE MONITORING UTILITY OF GLYCATED ALBUMIN AMONG DIABETIC PATIENTS ATTENDING CLINIC IN TERTIARY HOSPITALS IN DODOMA, TANZANIA: A CROSS-SECTIONAL STUDY PROTOCOL

PONE-D-23-20966R2

Dear Dr. Mkumbi,

We’re pleased to inform you that your manuscript has been judged scientifically suitable for publication and will be formally accepted for publication once it meets all outstanding technical requirements.

Kind regards,

Fredirick Lazaro mashili, MD, PhD

Academic Editor

PLOS ONE

Additional Editor Comments (optional):

The authors have sufficiently addressed all the comments raised by the reviewers.

Reviewers' comments:

Reviewer's Responses to Questions

**Comments to the Author**

1. Does the manuscript provide a valid rationale for the proposed study, with clearly identified and justified research questions?

Reviewer #3: Yes

2. Is the protocol technically sound and planned in a manner that will lead to a meaningful outcome and allow testing the stated hypotheses?

Reviewer #3: Yes

3. Is the methodology feasible and described in sufficient detail to allow the work to be replicable?

Reviewer #3: Yes

4. Have the authors described where all data underlying the findings will be made available when the study is complete?

Reviewer #3: Yes

5. Is the manuscript presented in an intelligible fashion and written in standard English?

Reviewer #3: Yes

6. Review Comments to the Author

You may also provide optional suggestions and comments to authors that they might find helpful in planning their study.

Reviewer #3: The manuscript is well written and the methodology is valid.

Nevertheless, the authors have responded that... "The six-month period has no meaning and was chosen arbitrarily just to exclude new and unstable diabetic patients from participating in the study while targeting stable anti-diabetic medication clients that have been at least on medication for some time and are assumed to have a set regimen that they adhere to. This aligns with the goal of the study to estimate glycemic control and suggest a better test, rather than to diagnose the disease."

Therefore, it is my opinion that such reasoning needs to be incorporated in the discussion section to enable the readers understand why the cut-off point for inclusion was set to be six months, and possibly state how this is likely to or not affect the study findings.

7. PLOS authors have the option to publish the peer review history of their article (what does this mean?). If published, this will include your full peer review and any attached files.

Reviewer #3: **Yes: **Alexander Mtemi Tungu
